# Main Fetal Predictors of Adverse Neonatal Outcomes in Pregnancies with Gestational Diabetes Mellitus

**DOI:** 10.3390/jcm9082409

**Published:** 2020-07-28

**Authors:** Maria-Christina Antoniou, Leah Gilbert, Justine Gross, Jean-Benoît Rossel, Céline Julie Fischer Fumeaux, Yvan Vial, Jardena Jacqueline Puder

**Affiliations:** 1Pediatric Service, Department Woman Mother Child, University Hospital of Lausanne, 1011 Lausanne, Switzerland; 2Obstetric Service, Department Woman Mother Child, University Hospital of Lausanne, 1011 Lausanne, Switzerland; leah.gilbert@chuv.ch (L.G.); justine.gross@chuv.ch (J.G.); jean-benoit.rossel@unisante.ch (J.-B.R.); yvan.vial@chuv.ch (Y.V.); jardena.puder@chuv.ch (J.J.P.); 3Service of Endocrinology, Diabetes and Metabolism, University Hospital of Lausanne, 1011 Lausanne, Switzerland; 4Clinic of Neonatology, Department Woman Mother Child, University Hospital of Lausanne, 1011 Lausanne, Switzerland; Celine-Julie.Fischer@chuv.ch

**Keywords:** gestational diabetes, fetal ultrasound, fetal anthropometry, pregnancy outcomes, neonatal complications

## Abstract

The objectives of this study were to (a) assess the utility of fetal anthropometric variables to predict the most relevant adverse neonatal outcomes in a treated population with gestational diabetes mellitus (GDM) beyond the known impact of maternal anthropometric and metabolic parameters and (b) to identify the most important fetal predictors. A total of 189 patients with GDM were included. The fetal predictors included sonographically assessed fetal weight centile (FWC), FWC > 90% and <10%, and fetal abdominal circumference centile (FACC), FACC > 90% and < 10%, at 29 0/7 to 35 6/7 weeks. Neonatal outcomes comprising neonatal weight centile (NWC), large and small for gestational age (LGA, SGA), hypoglycemia, prematurity, hospitalization for neonatal complication, and (emergency) cesarean section were evaluated. Regression analyses were conducted. Fetal variables predicted anthropometric neonatal outcomes, prematurity, cesarean section and emergency cesarean section. These associations were independent of maternal anthropometric and metabolic predictors, with the exception of cesarean section. FWC was the most significant predictor for NWC, LGA and SGA, while FACC was the most significant predictor for prematurity and FACC > 90% for emergency cesarean section. In women with GDM, third-trimester fetal anthropometric parameters have an important role in predicting adverse neonatal outcomes beyond the impact of maternal predictors.

## 1. Introduction

Gestational diabetes mellitus (GDM) carries an increased risk for short and long-term adverse outcomes, both for the mothers and their offspring [1,2]. In women with GDM, maternal anthropometric and metabolic parameters including prepregnancy BMI, gestational weight gain (GWG), maternal medical treatment requirement (metformin and/or insulin) and HbA1c at the end of pregnancy have been shown to influence and predict neonatal complications, such as small and large for gestational age (SGA, LGA), prematurity, hypoglycemia and cesarean section [3,4,5].

Third-trimester fetal ultrasound (US) is a helpful tool to predict neonatal outcomes. In the healthy pregnant population, estimated fetal weight (FW) was found to predict birth weight [6], whereas lower FW and fetal abdominal circumference (FAC) were associated with preterm birth [7,8]. There is a need for studies investigating the association between third-trimester fetal parameters and diverse adverse neonatal outcomes in the context of a population with GDM. Fetal abdominal circumference centile (FACC) cut-offs in association with maternal capillary glycemic values have been used for medical treatment guidance in women with GDM, leading to a reduction in neonatal complications, but these studies were limited to experienced centres and obstetricians [9,10,11].

To our knowledge, it is not known whether fetal anthropometric parameters have an added value in predicting diverse neonatal outcomes beyond the known impact of different maternal anthropometric and metabolic variables.

When analysing fetal US parameters to guide decisions for monitoring during pregnancy, the utility of each parameter can be assessed. Comparable efficiency between estimated FW and FACC has been shown to predict both SGA and LGA in a mixed population with diabetes [12]. To date, no study has compared the effectiveness of the fetal anthropometric parameters, including fetal weight centile (FWC), FACC and their lower and higher cut-offs, in predicting the most relevant neonatal outcomes in women with GDM.

To answer these questions, the objectives of this study were: (1) to assess the utility of fetal anthropometric parameters to predict the most relevant adverse neonatal outcomes beyond maternal anthropometric and metabolic parameters in a population of women with GDM and (2) to identify the most important fetal predictors for these outcomes.

## 2. Experimental Section

This is a prospective observational study, which included a consecutive cohort of pregnant women with GDM followed in the Diabetes and Pregnancy Unit in the Centre Hospitalier Universitaire Vaudois (CHUV), Lausanne, Switzerland, between April 2012 and October 2017. Detailed information on the material and methods have been described in a previous study [3]. Briefly, we included all women with GDM who had signed an informed consent. Exclusion criteria were: multiple gestation, pregestational diabetes or diabetes diagnosed before 13 weeks of gestation, missing newborn sex and/or birth weight and missing fetal ultrasound data between 29 0/7 and 35 6/7 gestational weeks. Patients with concomitant pathologies or pregnancy complications were not excluded from the study.

GDM was diagnosed according to the International Association of the Diabetes and Pregnancy Study Groups criteria [13], including a 75-g oral glucose tolerance test at 24–28 weeks GA. The treatment was based on the current guidelines of the American Diabetes Association [14] and of the Endocrine Society [15]. At their first clinical appointment, patients were seen by a diabetes educator specialized in GDM or a medical doctor, received information on GDM, and were taught how to perform the capillary blood glucose test. A dietician saw these women one week later and provided them with advice to optimal glycaemic control, while providing all the nutrients required and to promote optimal weight gain during pregnancy. Women were encouraged to increase physical activity and had the possibility to receive physical activity counselling by a physiotherapist, as well as to participate in GDM physical activity groups. According to international and local guidelines (Vaud Cantonal Diabetes Program [16,17]), women were asked to check their capillary glucose values 4x/day. If, despite lifestyle changes, glucose values remained above targets, metformin or insulin treatment was introduced [13,14,16,17].

### 2.1. Maternal and Fetal Predictors and Neonatal Outcome Measures

Maternal anthropometric and metabolic predictors included prepregnancy body mass index (BMI), GWG, fasting, 1-h and 2-h blood glucose values during the 75g oGTT at 24–28 weeks of GA, HbA1c at the last visit at the GDM clinic, and maternal glucose lowering medical treatment requirement (metformin and/or insulin). Prepregnancy BMI was calculated based on pre-pregnancy weight that was retrieved from medical charts or self-reported, and on height measured at the first visit at the GDM clinic, using the formula weight(kg)/(height(m))^2^. Height at the first GDM visit was measured to the nearest 0.1 cm with a regularly calibrated Seca^®^ height scale. GWG was determined as the difference between the last weight measured before delivery and pre-pregnancy weight. Weight was measured to the nearest 0.1 kg in women wearing light clothes and no shoes with an electronic scale (Seca^®^). HbA1c at the last visit at the GDM clinic (last visit before delivery) was performed after March 2015, and was measured using a chemical photometric method (conjugation with boronate; Afinion^®^). Maternal treatment was obtained from medical charts and classified into 2 categories (no treatment, treatment with metformin and/or insulin). The latter category was not subdivided, as only 14 women were treated with metformin alone.

Fetal predictors consisted of FWC (ranging from 0–100%), FWC > 90%, FWC < 10%, FACC (ranging from 0–100%), FACC > 90% and FACC < 10%. All fetal ultrasounds were performed at the CHUV by trained obstetricians. Estimated FW using the Hadlock formula [18] and FAC were obtained during the antenatal ultrasound performed between 29 0/7 and 35 6/7 weeks of gestation. Fetal centiles were calculated using the Intergrowth 21st fetal size application tool [19].

Neonatal outcomes included neonatal weight centile (NWC), large-for-gestational-age (LGA), small-for-gestational-age (SGA), hypoglycemia, prematurity, hospitalization in the neonatal unit for a neonatal complication, 5-min Apgar score < 7, cesarean section (emergency and scheduled together) and emergency cesarean section by itself. All neonatal outcomes, with the exception of NWC, were binary. Neonatal weight (g) was documented at birth as an absolute value; NWCs were calculated using the Intergrowth 21st newborn size application tool [20]. LGA was defined as newborn weight centile > 90% for sex and gestational age. SGA was defined as newborn weight centile < 10% for sex and gestational age. Prematurity was defined as gestational age < 37 weeks. Gestational age was calculated according to the date of the last menstruations, or as assessed by the fetal ultrasound in the cases where gestational age was corrected during the first trimester ultrasound evaluation. According to the centre protocol based on national Swiss guidelines [21], all neonates from mothers with GDM received feeding in the first 2 h of life and were fed every 2–3 h during the first 48 h in order to prevent neonatal hypoglycemia. Systemic blood glucose monitoring was conducted in all newborns [21], and the frequency of the controls depended on whether the mother was treated or not with insulin during her pregnancy (at least 3 controls, and at least 8 controls over 48 h in case of maternal treatment). Neonatal glycemia was also measured if symptoms suggested hypoglycemia. Neonatal hypoglycemia was defined as capillary or venous glucose value ≤ 2.5 mmol/L. The blood glucose value (capillary or venous) was also verified at the CHUV central laboratory, if capillary glycemia measured by the glucometer was ≤ 2.5 mmol/L. Neonates were hospitalized for intravenous glucose infusion when they presented a symptomatic hypoglycemia, or a glycemia ≤ 2.0 mmol/L, or more than one hypoglycemia ≤ 2.5 mmol/L despite administration of dextromaltan and/or formula milk. Any hospitalization in the neonatal unit was documented. Cesarean section occurrence was documented (emergency and total including also scheduled cesarean sections). Emergency cesarean section included all non-scheduled cesarean sections for either a fetal or maternal indication. The exact indication of the emergency cesarean section was not specified. In cases of scheduled cesarean section, the decision for the cesarean section indication was taken by the mother’s obstetrician as well the mother. Fetal and neonatal data were obtained from the center patient electronic medical chart for all newborns born in the CHUV.

### 2.2. Statistical Analysis

All data were analysed using Stata/SE 16.0 (StataCorp LLC, TX, USA). The normality of continuous variables was assessed, and normally distributed continuous variables were described as means and standard deviations (SDs). Binary outcomes were described as N (percentages) (Table 1). Linear and logistic regression analyses with adverse neonatal outcomes as the dependent variable, adjusting for gestational age at birth and neonatal sex where appropriate, and including the fetal and maternal variables as the predictor variables (see above), were initially conducted (Table A1 and Table A2 of the Appendix A). In the specific case of emergency cesarean, comparisons were made with scheduled cesarean section. In order to evaluate the role of fetal anthropometric parameters beyond maternal anthropometric and metabolic parameters, the analyses of significant fetal predictors were additionally adjusted for significant maternal predictors (Table 2). Finally, in order to identify the most important fetal predictors for neonatal outcomes, we used stepwise procedures (backward elimination), in order to select the most important variables among those which are highly correlated. (Table 3). This latter analysis was also adjusted for significant maternal predictors in univariate analyses, as well as for gestational age at birth and neonatal sex as indicated before. For all analyses, beta-coefficients (for continuous outcomes such as neonatal centiles) and adjusted odds ratios (aORs-for binary outcomes, e.g., all other neonatal outcomes) are reported along with their 95% confidence intervals (CIs). The significance was set at *p* < 0.05. Due to the small number of some neonatal complications, analysis was only performed for adverse outcomes present in more than 10 cases [22]. Therefore, 5-min Apgar score < 7 was removed from the regression analyses.

Table 3 shows the main fetal predictors of neonatal outcomes using multiple logistic regression analyses. FWC was the most relevant fetal predictor for NWC, LGA, and SGA (inverse association; all *p* < 0.001). FACC was the most relevant predictor for prematurity (inverse association) and FACC > 90% for emergency cesarean section (both *p* ≤ 0.029).

### 2.3. Ethics

Signed informed consent was obtained from all participating women. The study was conducted in accordance with the guidelines of the declaration of Helsinki, and good clinical practice. The Human Research Ethics Committee of the Canton de Vaud approved the study protocol (326/15).

## 3. Results

Out of a population of 826 adult women with gestational diabetes, 111 women were excluded due to missing informed consent, 9 because they participated in an intervention clinical trial, 128 due to multiple gestation, missing newborn sex and/or birth weight and 389 because of missing fetal ultrasound data between 29 0/7 and 35 6/7 gestational weeks. Overall, 189 women were included in the final analysis.

### 3.1. Maternal, Fetal and Neonatal Characteristics

Detailed information about the maternal, fetal and neonatal characteristics are shown in Table 1. In summary, women were 32.9 ± 5.4 years old and had a mean prepregnancy BMI of 26.6 ± 5.4 kg/m2. The mean fetal and neonatal weight centiles were 67.8 ± 21.4%, and 55.3 ± 31.7%, respectively.

### 3.2. Associations Between Maternal and Fetal Predictors and Neonatal Outcomes

Prepregnancy maternal BMI, GWG, fasting, 1h and 2h glucose values at oGTT and need for maternal glucose lowering medical treatment (metformin and/or insulin) showed a significant association with one or more adverse neonatal outcomes such as NWC, LGA, SGA, hypoglycemia, and cesarean section (all *p* ≤ 0.046, see Table A1 of the Appendix A). HbA1c at the last GDM visit did not show any association with neonatal outcomes. The maternal medical treatment requirement was the only maternal predictor for neonatal hypoglycemia (*p* = 0.02), whereas none of the maternal parameters were correlated with prematurity, hospitalization for neonatal complications or emergency cesarean section.

One or more of the fetal parameters were correlated with all adverse neonatal outcomes except for hypoglycemia and hospitalization for neonatal complications (all *p* ≤ 0.047, see Table A2 of the Appendix A). The significance of fetal predictors did not change after adjusting for significant maternal predictors, with the exception of cesarean section; therefore, the FACC < 10% did not remain significant after adjustment for maternal glycemic values at the oGTT (Table A2 of the Appendix A and Table 2). Thus, after adjustment for maternal predictors, FWC was positively associated with NWC, LGA, and inversely with SGA and prematurity (all *p* ≤ 0.038), while FWC > 90% showed a positive correlation with neonatal birth weight, LGA, and emergency cesarean section (all *p* ≤ 0.047). Similarly, FACC was positively associated with neonatal birth weight, LGA, and inversely with SGA and prematurity (all *p* ≤ 0.029). FACC > 90% showed a positive association with neonatal birth weight, LGA and emergency cesarean section, and an inverse association with SGA (all *p* ≤ 0.049).

## 4. Discussion

The novel finding in this study of 189 clinically followed women with GDM was that third-trimester fetal anthropometric parameters can predict diverse relevant neonatal outcomes, such as anthropometry, prematurity, and emergency cesarean section, independently and beyond the impact of significant maternal anthropometric and metabolic predictors. However, none of the maternal or fetal parameters could predict hospitalization for neonatal complications. Maternal glucose lowering medical treatment requirement was the only predictor for neonatal hypoglycemia. FWC was found to be the most powerful predictor for NWC, LGA, and SGA, whereas FACC was the most powerful predictor for prematurity, and FACC > 90% for emergency cesarean section. Based on our findings, fetal ultrasound is a useful tool in the management of women with GDM, helping to independently predict adverse neonatal outcomes.

More specifically, FWC was an independent predictor for NWC, LGA, SGA and prematurity, while FWC > 90% was an independent predictor for neonatal birth weight, LGA, and emergency cesarean section. The association between FW and neonatal complications in the context of a population with GDM in a clinical setting still remains poorly studied, even in healthy pregnancies, studies mainly focused on anthropometric neonatal outcomes or a single adverse outcome. A previous study in a healthy population showed that estimated FW was a reliable predictor of actual birth weight; sonography appeared marginally more accurate in predicting SGA than macrosomia [6]. Alsulyman et al. compared discrepancies between intrapartum sonographically estimated FW and actual birth weight, and found a similar accuracy between women with (mixed GDM and pre-existent diabetes) and without diabetes [23]. Estimated FW assessed by ultrasound between 36 0/7 and 38 6/7 weeks of gestation predicted emergency cesarean section in a recent retrospective study including a predominantly healthy population (18% GDM) [24]. To our knowledge, this is the first study proving the utility of third-trimester sonographically estimated FW in the prediction of a series of neonatal outcomes, in the context of a population with GDM.

Moreover, FACC (%) independently predicted neonatal birth weight, LGA, SGA and prematurity, while FACC > 90% predicted neonatal birth weight, LGA, SGA and emergency cesarean section. Our study is in accordance with a study by Hawkings et al., which showed that in a healthy population, FACC < 10% was associated with a higher incidence of preterm delivery. Previous studies have used different sonographically assessed FACC cut-offs (>70% or 75% centile) and maternal capillary glycemic values in order to guide the medical treatment in populations with GDM [9,10,11]. GDM management based on FACC cut-offs, combined with less stringent glycemic criteria, resulted in similar rates of cesarean section, LGA, SGA, neonatal hypoglycemia, and neonatal admission compared to management based on strict glycemic criteria alone in a study by Schaefer-Graf et al. [10] and in lower rates of LGA, macrosomia and SGA in a study by Bonomo et al. [11]. A third-trimester FACC cut-off could be used for treatment guidance in women with GDM, aiming to reduce adverse neonatal outcomes.

We also evaluated the respective importance of FWC and FACCs, and their higher and lower cut-offs in predicting neonatal outcomes. FWC was found to be the most powerful fetal predictor for NWC, LGA, and SGA, while FACC was superior for the prediction of prematurity, and FACC >90% was a stronger predictor for emergency cesarean section. To our knowledge, this is the first study comparing the role of different fetal anthropometric parameters in the prediction of a series of neonatal outcomes in the context of a population with GDM. A previous study by Holcomb et al. demonstrated equal efficiency between sonographically estimated FW and FAC (without using centiles or cut-offs) in the prediction LGA, in a mixed population with diabetes [12]. In our study, FWC was more relevant for the prediction of neonatal anthropometric parameters at birth, whereas FACC was more relevant for the prediction of other outcomes, such as prematurity and emergency cesarean section. Thus, both parameters are useful and non-interchangeable in the follow-up of patients with GDM. In terms of clinical relevance, these parameters may be implemented in clinical practice for maternal treatment guidance, enabling a personalized treatment based on maternal metabolic control and fetal anthropometry.

HbA1c at the last visit at the GDM clinic was not associated with neonatal outcomes, which may be related to a smaller sample size due to missing data.

The strengths of our study included its originality and prospective nature, which ensured the presence of complete detailed information on maternal, fetal and neonatal characteristics. However, some limitations may also be noted. The emergency cesarean section indication was not specified, and the premature population was included as a whole. Dividing the population into emergency cesarean section subgroups (i.e., for maternal or fetal reason), as well as the subclassification of prematurity according to gestational age at delivery, could be interesting, but would lead to smaller sizes and limited statistical power. Moreover, as the exact indication for emergency cesarean was not specified, some of these may have been due to pregnancy complications not directly connected with gestational diabetes. The presence of an instrumentally assisted vaginal birth was also not investigated, due to insufficiently documented data. Lastly, we did not include fetal anthropometric data obtained during the second trimester of pregnancy, due to the limited number of patients followed at our tertiary hospital before the diagnosis of GDM. This was the authors’ decision in order to ensure that all ultrasound measurements were performed with the same methodology and by an equally experienced team, in order to ensure data quality.

## 5. Conclusions

FWC and FACCs assessed in the third trimester predicted diverse relevant adverse neonatal outcomes at birth independently and beyond the impact of maternal anthropometric and metabolic parameters. FWC was found to be the most relevant predictor for neonatal anthropometric parameters (weight centile, LGA, SGA), and FACC and FACC > 90% were the most relevant predictors for prematurity and emergency cesarean section, respectively. Fetal anthropometry is thus a useful tool for risk stratification in pregnancies with GDM. Along with maternal anthropometric and metabolic parameters such as weight (changes) and glycemic control, it could be used for maternal treatment guidance, allowing for a personalized follow-up and eventually a decrease in adverse neonatal outcomes.

## 6. Patents

Not applicable.

## Figures and Tables

**Table 1 jcm-09-02409-t001:** Descriptive maternal, fetal and neonatal characteristics.

Number of Patients	189
Maternal characteristics	
Age (years)	32.9 ± 5.4
Prepregnancy BMI (kg/m^2^)	26.6 ± 5.4
Gestational weight gain (kg)	13.3 ± 7.2
Gestational weight gain until the 1st visit at the GDM clinic (kg)	10.5 ± 6.1
Fasting oGTT glucose value (mmol/L)	5.3 ± 0.7
1-h oGTT glucose value (mmol/L)	10.0 ± 2.1
2-h oGTT glucose value (mmol/L)	7.9 ± 2.0
Gestational age at the 1st visit at the GDM clinic (weeks)	28.2 ± 3.0
Gestational age at the last visit at the GDM clinic *(weeks)	36.1 ± 1.4
HbA1c at the last visit at the GDM clinic * (%)	5.7 ± 0.5
Maternal medical treatment requirement N(%)	104 (58.8)
Fetal characteristics	
Gestational age (weeks)	32.8 ± 1.5
Fetal weight centile * (%)	67.8 ± 21.4
Fetal weight centile > 90% * N(%)	33 (17.5)
Fetal weight centile < 10%* N(%)	0
Fetal abdominal circumference centile * (%)	65.9 ± 29.9
Fetal abdominal circumference centile > 90% * N(%)	54 (28.6)
Fetal abdominal circumference centile < 10% * N(%)	12 (6.4)
Neonatal characteristics	
Male N(%)	96 (50.8)
Gestational age at birth (weeks)	38.8 ± 1.5
Neonatal weight (g)	3252 ± 591
Neonatal weight centile ^†^ (%)	55.3 ± 31.7
LGA ^‡^ N(%)	41 (21.7)
SGA ^§^ N(%)	22 (11.6)
Neonatal Hypoglycemia ** N(%)	25 (13.9)
Prematurity ^††^ N(%)	16 (8.5)
Hospitalization for neonatal complication N(%)	25 (13.8)
5-min Apgar score < 7 N(%)	6 (3.2)
Cesarean section ^‡‡^ N(%)	88 (48.1)
Emergency cesarean section N(%)	41 (22.4)

Abbreviations: BMI body mass index, GDM gestational diabetes mellitus, oGTT oral glucose tolerance test, HbA1c glycated hemoglobin, LGA Large for gestational age, SGA Small for gestational age. * for gestational age using the Intergrowth 21st fetal size application tool [19] ^†^ for sex and gestational age using the Intergrowth 21st newborn size application tool [20]. ^‡^ LGA: birth weight >90th centile for sex and gestational age using the Intergrowth 21st newborn size application tool [20]. ^§^ SGA: birth weight < 10th centile for sex and gestational age using the Intergrowth 21st newborn size application tool [20]. ** capillary or venous glucose value ≤ 2.5 mmol/L. ^††^ gestational age <37 weeks. ^‡‡^ cesarean section includes scheduled and emergency cesarean sections.

**Table 2 jcm-09-02409-t002:** Fetal predictors of adverse neonatal outcomes after adjustment for maternal predictors.

Neonatal Outcomes	Fetal Predictors	OR/ Beta-Coefficient	Standard Error	95% CI		*p* Value
Neonatal Weight Centile (%) *	Fetal weight centile (%) ^†^	0.94 ^††^	0.09	0.76	1.13	<0.001
	Fetal weight centile >90 (%) ^†^	37.77 ^††^	5.62	22.66	44.88	<0.001
	Fetal abdominal circumference centile (%) ^†^	0.55 ^††^	0.07	0.40	0.69	<0.001
	Fetal abdominal circumference centile >90 (%) ^†^	30.27 ^††^	4.78	20.83	39.71	<0.001
LGA ^‡^	Fetal weight centile (%) ^†^	1.09	0.03	1.04	1.14	<0.001
	Fetal weight centile >90 (%) ^†^	10.90	7.10	3.04	39.09	<0.001
	Fetal abdominal circumference centile (%) ^†^	1.09	0.03	1.03	1.14	0.001
	Fetal abdominal circumference centile >90 (%) ^†^	9.46	5.54	3.00	29.83	<0.001
SGA ^§^	Fetal weight centile (%) ^†^	0.95	0.01	0.92	0.97	<0.001
	Fetal abdominal circumference centile (%) ^†^	0.97	0.01	0.96	0.99	<0.001
	Fetal abdominal circumference centile >90 (%) ^†^	0.13	0.13	0.02	0.99	0.049
Prematurity ^¶^	Fetal weight centile (%) ^†^	0.98	0.01	0.95	1.00	0.038
	Fetal abdominal circumference centile (%) ^†^	0.98	0.01	0.97	1.00	0.029
Cesarean section **	Fetal abdominal circumference centile < 10 (%) ^†^	0.38	0.35	0.07	2.27	0.291
Emergency cesarean section **	Fetal weight centile > 90 (%) ^†^	3.08	1.75	1.01	9.38	0.047
	Fetal abdominal circumference centile > 90 (%) ^†^	3.17	1.58	1.20	8.41	0.020

Abbreviations: OR odds ratio BMI body mass index, GDM gestational diabetes mellitus, oGTT oral glucose tolerance test, HbA1c glycated hemoglobin, LGA Large for gestational age, SGA Small for gestational age. * for sex and gestational age using the Intergrowth 21st newborn size application tool [20]. ^†^ for gestational age using the Intergrowth 21st fetal size application tool [19]. ^‡^ LGA: birth weight >90th centile for sex and gestational age using the Intergrowth 21st newborn size application tool [20]. ^§^ SGA: birth weight <10th centile for sex and gestational age using the Intergrowth 21st newborn size application tool [20]. ^¶^ gestational age < 37 weeks. ** cesarean section includes scheduled and emergency cesarean sections. Emergency cesarean sections were compared to scheduled cesarean sections. ^††^ this value corresponds to a beta-coefficient. Linear and logistic regression analyses, adjusted for neonatal sex, gestational age and significant maternal variables presented in Table A1.

**Table 3 jcm-09-02409-t003:** Main fetal predictors of adverse neonatal and maternal outcomes.

Neonatal Outcomes	Fetal Predictors	OR/ Beta-Coefficient	Standard Error	95% CI		*p* Value
Neonatal Weight centile (%) *	Fetal weight centile (%) ^†^	0.94 ^††^	0.09	0.76	1.13	<0.001
	Fetal weight centile > 90(%) ^†^					0.306
	Fetal abdominal circumference centile (%) ^†^					0.616
	Fetal abdominal circumference centile > 90 (%) ^†^					0.887
LGA ^‡^	Fetal weight centile (%) ^†^	1.09	0.260	1.04	1.14	<0.001
	Fetal weight centile > 90 (%) ^†^					0.569
	Fetal abdominal circumference centile (%) ^†^					0.287
	Fetal abdominal circumference centile > 90 (%) ^†^					0.937
SGA ^§^	Fetal weight centile (%) ^†^	0.950	0.1200	0.920	0.970	<0.001
	Fetal abdominal circumference centile (%) ^†^					0.750
	Fetal abdominal circumference centile > 90 (%) ^†^					0.829
Prematurity ^¶^	Fetal weight centile (%) ^†^					0.634
	Fetal abdominal circumference centile (%) ^†^	0.98	0.01	0.97	1.00	0.029
Emergency cesarean section **	Fetal weight centile >90 (%) ^†^					0.756
	Fetal abdominal circumference centile >90 (%) ^†^	1.15	0.50	0.18	2.13	0.020

Abbreviations: OR odds ratio BMI body mass index, GDM gestational diabetes mellitus, oGTT oral glucose tolerance test, HbA1c glycated hemoglobin, LGA Large for gestational age, SGA Small for gestational age. * for sex and gestational age using the Intergrowth 21st newborn size application tool [20]. ^†^ for gestational age using the Intergrowth 21st fetal size application tool [19]. ^‡^ LGA: birth weight >90th centile for sex and gestational age using the Intergrowth 21st newborn size application tool [20]. ^§^ SGA: birth weight <10th centile for sex and gestational age using the Intergrowth 21st newborn size application tool [20]. ^¶^ gestational age < 37 weeks. ** emergency cesarean section was compared to scheduled cesarean section. ^††^ this value corresponds to a beta-coefficient. Manual stepwise multiple logistic regression analyses with all significant fetal variables presented in Table A2, adjusted for neonatal sex and gestational age, as well as significant maternal variables presented in Table A1. The outcomes are only shown if at least one predictor remains significative.

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
