# Peer review of "Main Fetal Predictors of Adverse Neonatal Outcomes in Pregnancies with Gestational Diabetes Mellitus"

_jcm, 2020, doi:10.3390/jcm9082409_

Round 1
Reviewer 1 Report
Gestational diabetes mellitus (GDM) is associated with increased risk of maternal complications, adverse pregnancy and neonatal outcomes (macrosomy, preeclampsia, intrauterine growth restriction, neonatal hypoglycemia, hyperbilirubinemia, hypocalcemia, respiratory distress syndrome, polycythemia). Clinical prediction of these complications gives the possibility to perform preventative and therapeutic interventions. The most important problem is a prediction of large for gestational age newborns (LGA) as a cause of failure to progress in labour, obstetric intervention and shoulder dystocia.
In clinical practice the introduction of a clinical model with widely available clinical factors may be very useful to identify those pregnancies that are at increased risk of complications.
The aim of this study was to assess the utility of fetal anthropometric variables for prediction the most relevant adverse neonatal outcomes in GDM patients in addition to the known impact of maternal anthropometric and metabolic parameters and to identify the most important fetal predictors. The main conclusion is that in GDM women third trimester fetal anthropometric parameters have an important role in predicting adverse neonatal outcomes beyond the impact of maternal predictors.
This study is technically very well-performed and the conclusions are interesting and informative.
In my opinion some points should be additionally explained or changed:
- Line 29: AC centile (<10%)
- Line 101: cesarean section (emergency and scheduled) and emergency cesarean section?
- Table 1.: Maternal characteristic: gestational age at the end of pregnancy 36.1±1.4 but in neonatal characteristics: gestational age (weeks) 38.8?! ±1.5. What was the time of delivery?
- Table 1.: Fetal characteristics: fetal weight centile <10% - 0. In the population of 189 patient? There are no data about inclusion criteria for the study. Patients with the other pregnancy complications: preeclampsia, metabolic syndrome, chronic renal and liver diseases, chronic infectious diseases, systemic lupus erythematosus … with the higher risk of intrauterine growth restriction were excluded from the study?
- As authors described, the emergency cesarean section indication was not specified because dividing the population in emergency cesarean section subgroups could lead to smaller sizes and limited statistical power. But some of the emergency cesarean section indications are completely not connected with GDM, fetal weight and abdominal circumference of the fetus but with other pregnancy complications. It should be explained (also inclusion criteria).
- According to the reference number 24 the risk for cesarean section is doubled if the AC is above the 75th centile. The prediction of large for gestational age newborns as a cause of failure to progress in labour, shoulder dystocia and emergency cesarean section is very important. What was the approach in this study? When AC during ultrasound examination between 29 0/7 and 35 6/7 was >90%, the next examination was performed before delivery for preventing complications and elective (not emergency) cesarean section was performed in macrosomic/high AC fetuses? It should be explained.
Author Response
Dear reviewer,
Thank you for your helpful comments that helped to clarify certain parts and to improve the current manuscript. We adapted our manuscript to the comments. Below you find our point-to point review.
In our responses, the lines correspond to the lines in the revised manuscript.
We hope, the manuscript is now acceptable in its current form.
Yours sincerely,
The authors
Maria-Christina Antoniou
Leah Gilbert
Justine Gross
Jean-Benoît Rossel
Céline J. Fischer Fumeaux
Yvan Vial
Jardena J. Puder
Reviewer 1:
1)Line 29: AC centile (<10%)
LINE 29: In line 29, it was AC centile (not AC centile <10%) was found to be the most significant predictor for prematurity
2)Line 101: cesarean section (emergency and scheduled) and emergency cesarean section?
LINES 114-115: Thank you for this comment. Cesarean section was one of the outcomes evaluated in this study. It was divided in overall cesarean section, including both emergency and scheduled cesarean section as well as emergency cesarean section by itself as its health impact is especially pronounced. We specified this now in the manuscript.
3) Table 1.: Maternal characteristic: gestational age at the end of pregnancy 36.1±1.4 but in neonatal characteristics: gestational age (weeks) 38.8?! ±1.5. What was the time of delivery?
Table 1.: Gestational age at the end of pregnancy corresponds to the age at the last visit at the GDM clinic, whereas gestational age corresponds to the age at birth. Thanks to this comment, we now made the tables and manuscript clearer and replaced “at the end of pregnancy”, by “at the last visit at the GDM clinic”. We also replaced “Gestational age” by “Gestational age at birth” in Table 1.
4) Table 1.: Fetal characteristics: fetal weight centile <10% - 0. In the population of 189 patient? There are no data about inclusion criteria for the study. Patients with the other pregnancy complications: preeclampsia, metabolic syndrome, chronic renal and liver diseases, chronic infectious diseases, systemic lupus erythematosus … with the higher risk of intrauterine growth restriction were excluded from the study?
Table 1.: We were also surprised by the absence of fetuses with a weight centile <10%. However, small for gestational age prevalence at birth was 11.6%. We did not exclude any patients for any pregnancy complications and clarified this now in the experimental section (lines 70-75 “Briefly, we included all women with GDM who had signed an informed consent. Exclusion criteria were: multiple gestation, pregestational diabetes or diabetes diagnosed before 13 weeks of gestation, missing newborn sex and/or birth weight and missing fetal ultrasound data between 29 0/7 and 35 6/7 gestational weeks. Patients with concomitant pathologies or pregnancy complications were not excluded from the study.
.”).
5) As authors described, the emergency cesarean section indication was not specified because dividing the population in emergency cesarean section subgroups could lead to smaller sizes and limited statistical power. But some of the emergency cesarean section indications are completely not connected with GDM, fetal weight and abdominal circumference of the fetus but with other pregnancy complications. It should be explained (also inclusion criteria).
Thank you for addressing this interesting point. Unfortunately, the emergency cesarean section indication was not specified, and some emergency cesarean sections may have been due to pregnancy complications not directly linked to GDM. A phrase was added in the experimental section and in the discussion section as a limitation (line 136 “The exact indication of the emergency cesarean section was not specified.”).
6) According to the reference number 24 the risk for cesarean section is doubled if the AC is above the 75th centile. The prediction of large for gestational age newborns as a cause of failure to progress in labour, shoulder dystocia and emergency cesarean section is very important. What was the approach in this study? When AC during ultrasound examination between 29 0/7 and 35 6/7 was >90%, the next examination was performed before delivery for preventing complications and elective (not emergency) cesarean section was performed in macrosomic/high AC fetuses? It should be explained.
Thank you for this question. According to the current approach of the obstetric team of the CHUV, in the cases where abdominal circumference centile is ≥ 90% at the ultrasound examination between 29 0/7 and 35 6/7, an ultrasound is conducted at 38 gestational weeks and if the abdominal circumference centile or the estimated weight is ≥ 90%, an induction of labor is proposed between 38 and 39 gestational weeks.
Reviewer 2 Report
The authors have studied the fetal predictors of adverse neonatal outcome in pregnancies in gestational diabetes mellitus. The study is well designed and the results are presented clearly. In the result section, the authors have said why they did not include women with certain conditions. Since it is not relevant to the study, they can take it off and instead, give more details about the parameters used for the study
Author Response
Dear reviewer,
Thank you for your helpful comments that helped to clarify certain parts and to improve the current manuscript. We adapted our manuscript to the comments.
We hope, the manuscript is now acceptable in its current form.
Yours sincerely,
The authors
Maria-Christina Antoniou
Leah Gilbert
Justine Gross
Jean-Benoît Rossel
Céline J. Fischer Fumeaux
Yvan Vial
Jardena J. Puder
The authors have studied the fetal predictors of adverse neonatal outcome in pregnancies in gestational diabetes mellitus. The study is well designed and the results are presented clearly. In the result section, the authors have said why they did not include women with certain conditions. Since it is not relevant to the study, they can take it off and instead, give more details about the parameters used for the study
Thank you for your favorable review. Unfortunately, the 2 other reviewers asked for more precise detail on the population, the inclusion and exclusion criteria of our study, so we were obliged to add some information in the experimental part and discussion. In response to this reviewer’s comment, we have also added some details on some parameters used in the study in the experimental section (lines 96-97 “Height at the first GDM visit was measured to the nearest 0.1 cm with a regularly calibrated Seca® height scale.” and lines 98-99 “Weight was measured to the nearest 0.1 kg in women wearing light clothes and no shoes with an electronic scale (Seca®).”).
Reviewer 3 Report
Thank you for the opportunity to review your manuscript. The following suggested edits/questions/comments are offered with all due respect to the authors.
ABSTRACT
LINE 17: It would be helpful to spell out “GDM” the first time it is referred to, i.e., gestational diabetes mellitus (GDM)
LINES 17-19: Did you mean, "assess the utility of fetal anthropometric variables to predict the most relevant adverse neonatal outcomes in a treated population with GDM AS WELL AS TO ASSESS the known impact of maternal anthropometric and metabolic parameters...”?
LINES 21-22, 93-95: Need for clarification. Did you mean: “...fetal weight (FW) centile (>90% and <10%) and abdominal circumference (AC) centile (>90% and <10%)”?
INTRODUCTION
LINES 39, 84: The term "maternal treatment requirement” is vague. Is this referring to medical treatment requirements for women with GDM? In Lines 90-91, clarification is provided (i.e., that this refers to the GDM management regimen). It would help to clarify this earlier.
LINE 53: "When analysing fetal US parameters to guide decisions for monitoring...” Monitoring the pregnancy or blood glucose? I suspect it’s the former.
LINES 55, 262, 263, 290: Please avoid use of the term “diabetic”/“prediabetic" when referring to people with diabetes or pre-diabetes. See Australia and American Diabetes Association diabetes language guidelines for more information.
EXPERIMENTAL SECTION
LINES 73-74: Did you mean, "A dietician saw these women one week later and provided them with advice to OPTIMAL glycaemic control...”?
LINES 78-79: Did you mean, “...women were asked to CHECK their capillary glucose values 4x/day”?
LINE 100: “hypoglycemia” This is also listed in Lines 23 and 41. Is this referring to the presence of hypoglycemia as a dichotomous variable (i.e., yes/no)? [Also, the comma is missing between SGA and hypoglycemia in Line 100.] Starting with Line 110, the term "neonatal hypoglycemia” is used and may be helpful to use, as applicable, throughout the manuscript for clarity.
LINES 101-102: It’s not necessary to list "cesarean section (emergency and scheduled)" and "emergency cesarean section” separately. It’s probably sufficient to state "cesarean section (emergency and scheduled)” [Refer to Lines 119-120, "Cesarean section occurrence was documented (emergency and total including also scheduled cesarean sections).”]
RESULTS
*What about HbA1c data? Given that this is such a widely used measurement, it would be helpful to address this variable. Thought: If HbA1c was not a predictive variable, might time-in-range and the use of continuous glucose monitors serve a useful purpose in women with GDM?
LINE 167: Is "need for medical treatment” referring to GDM-related treatment or any pregnancy-related treatment? I suspect GDM-related. If so, perhaps the term "GDM-related treatment" can be used?
LINE 169: Is "Maternal medical treatment requirement” referring to GDM-related treatment or any pregnancy-related treatment? I suspect GDM-related. If so, perhaps the term "GDM-related treatment requirement" can be used?
LINE 180: It would be helpful to use “FW” consistently (instead of spelling out fetal weight) throughout the manuscript.
LINES 179, 182, 184, 189, 190: Are the inverse associations indicated in Table 2 and Table 3? I believe they are reflected in Table A2.
DISCUSSION
*What about HbA1c data?
LINES 248-249: Is "Maternal medical treatment” referring to GDM-related treatment or any pregnancy-related treatment? I suspect GDM-related. If so, perhaps the term "GDM-related treatment" can be used?
LINES 249-252: Did you mean, “... whereas abdominal 250 circumference centile WAS THE MOST POWERFUL PREDICTOR for prematurity, and abdominal circumference centile >90% for emergency 251 cesarean section”?
LINES 254-256: Did you mean, “...while fetal weight centile > 90% WAS AN INDEPENDENT PREDICTOR for neonatal birth weight, LGA, and 255 emergency cesarean section”?
LINES 257, 267, “...a treated population...” Is this referring to women with GDM who managed their blood glucose with medication?
LINES 283-286: Did you mean, "Fetal weight centile was found to be the most powerful fetal predictor for neonatal weight centile, LGA, and SGA, while abdominal circumference centile was superior for the prediction of prematurity AND abdominal circumference centile >90% WAS A STRONGER PREDICTOR for emergency cesarean section”?
LINES 291-293: Did you mean, "In our study, fetal weight centile was more relevant for the prediction of neonatal anthropometric parameters at birth, whereas abdominal circumference WAS MORE RELEVANT for the prediction of other outcomes such as prematurity and emergency cesarean section."
CONCLUSIONS
*What about HbA1c data?
LINES 312-315: Did you mean, "Fetal weight centile was found to be the most relevant predictor for neonatal anthropometric parameters (weight centile, LGA, SGA), WHEREAS abdominal circumference centile and abdominal circumference centile >90% WERE THE MOST RELEVANT PREDICTORS for prematurity and emergency cesarean section respectively."
Author Response
Dear reviewer,
Thank you for your helpful comments that helped to clarify certain parts and to improve the current manuscript. We adapted our manuscript to the comments. Below you find our point-to point review.
In our responses, the lines correspond to the lines in the revised manuscript.
We hope, the manuscript is now acceptable in its current form.
Yours sincerely,
The authors
Maria-Christina Antoniou
Leah Gilbert
Justine Gross
Jean-Benoît Rossel
Céline J. Fischer Fumeaux
Yvan Vial
Jardena J. Puder
ABSTRACT
1) LINE 17: It would be helpful to spell out “GDM” the first time it is referred to, i.e., gestational diabetes mellitus (GDM)
LINE18-19: As proposed, in the latest version of the manuscript, “GDM” was substituted by gestational diabetes mellitus (GDM).
2) LINES 17-19: Did you mean, "assess the utility of fetal anthropometric variables to predict the most relevant adverse neonatal outcomes in a treated population with GDM AS WELL AS TO ASSESS the known impact of maternal anthropometric and metabolic parameters...”?
LINES 17-20: Thank you for this constructive comment that helped us to clarify the aims in the abstract. The aim was not to assess the known impact of maternal parameters, but the utility of fetal variables beyond the known impact of maternal variables. We now adjusted this in the manuscript: “The objectives of this study were to a) assess the utility of fetal anthropometric variables to predict the most relevant adverse neonatal outcomes in a treated population with gestational diabetes mellitus (GDM) beyond the known impact of maternal anthropometric and metabolic parameters …”
3) LINES 21-22, 93-95: Need for clarification. Did you mean: “...fetal weight (FW) centile (>90% and <10%) and abdominal circumference (AC) centile (>90% and <10%)”?
Our study included 6 fetal predictors 1) fetal weight centile (FWC), 2) FWC >90%, 3) FWC <10%, 4) fetal abdominal circumference centile (FACC), 5) FACC >90% and 6) FACC <10%. In the revised manuscript these abbreviations were added in the whole manuscript as detailed in the answer 13.
INTRODUCTION
4) LINES 39, 84: The term "maternal treatment requirement” is vague. Is this referring to medical treatment requirements for women with GDM? In Lines 90-91, clarification is provided (i.e., that this refers to the GDM management regimen). It would help to clarify this earlier.
LINES 40, 94: Thank you for this comment. This point was clarified in the manuscript. More precisely: line 40: “maternal medical treatment requirement (metformin and/or insulin)” and line 94: “maternal glucose lowering medical treatment requirement (metformin and/or insulin).”
5) LINE 53: "When analysing fetal US parameters to guide decisions for monitoring...” Monitoring the pregnancy or blood glucose? I suspect it’s the former.
LINES 55-56: This point was clarified in the manuscript. More precisely: “When analysing fetal US parameters to guide decisions for monitoring during pregnancy, the utility of each parameter can be assessed …”
6) LINES 55, 262, 263, 290: Please avoid use of the term “diabetic”/“prediabetic" when referring to people with diabetes or pre-diabetes. See Australia and American Diabetes Association diabetes language guidelines for more information.
LINES 58, 284-286, 314: Thank you for this useful comment. The text was adapted accordingly: line 56: “a mixed population with diabetes”, line 284-286: “…and found a similar accuracy between women with (mixed GDM and pre-existent diabetes) and without diabetes.”, line 314: “In a mixed population with diabetes”
EXPERIMENTAL SECTION
7) LINES 73-74: Did you mean, "A dietician saw these women one week later and provided them with advice to OPTIMAL glycaemic control...”?
LINES 81-82: The sentence was adapted according to your proposition: “A dietician saw these women one week later and provided them with advice to optimal glycaemic control”,
8) LINES 78-79: Did you mean, “...women were asked to CHECK their capillary glucose values 4x/day”?
LINES 85-87: The sentence was adapted according to your proposition: “According to international and local guidelines (Vaud Cantonal Diabetes Program [16, 17]), women were asked to check their capillary glucose values 4x/day.”
9) LINE 100: “hypoglycemia” This is also listed in Lines 23 and 41. Is this referring to the presence of hypoglycemia as a dichotomous variable (i.e., yes/no)? [Also, the comma is missing between SGA and hypoglycemia in Line 100.] Starting with Line 110, the term "neonatal hypoglycemia” is used and may be helpful to use, as applicable, throughout the manuscript for clarity.
LINE 114: A comma was added between (SGA) and hypoglycemia.
LINE 116-117, 119, Table 1, Table A1 and Table A2 of the appendix: Thank you for this comment. The sentence “All neonatal outcomes, with the exception of neonatal weight centile were binary” was added. We also replaced “hypoglycemia” by "neonatal hypoglycemia” in the experimental section (line 129), in Table 1, Table A1 and Table A2 of the appendix as requested for clarity.
10) LINES 101-102: It’s not necessary to list "cesarean section (emergency and scheduled)" and "emergency cesarean section” separately. It’s probably sufficient to state "cesarean section (emergency and scheduled)” [Refer to Lines 119-120, "Cesarean section occurrence was documented (emergency and total including also scheduled cesarean sections).”]
As reviewer 1 asked us to even more specify the outcome “overall cesarean section” vs “emergency cesarean section” (with a specifically pronounced impact on health outcomes), we left both outcomes.
RESULTS
11) *What about HbA1c data? Given that this is such a widely used measurement, it would be helpful to address this variable. Thought: If HbA1c was not a predictive variable, might time-in-range and the use of continuous glucose monitors serve a useful purpose in women with GDM?
HbA1c at the end of pregnancy was not found to be a significant predictor of any neonatal outcome (Table A1 of the Appendix). The sentence “HbA1c at the last GDM visit did not show any association with neonatal outcomes.” was added in the results section (Lines 186-187). Your idea of time in range is possibly very true. As we focus on fetal predictors (and maternal predictors are rather covariates), we propose not to add these options and perspectives in this current manuscript.
12) LINE 167: Is "need for medical treatment” referring to GDM-related treatment or any pregnancy-related treatment? I suspect GDM-related. If so, perhaps the term "GDM-related treatment" can be used?
LINE 169: Is "Maternal medical treatment requirement” referring to GDM-related treatment or any pregnancy-related treatment? I suspect GDM-related. If so, perhaps the term "GDM-related treatment requirement" can be used?
LINES 183-184: Thank you for this comment. The need for maternal treatment refers only to GDM-related treatment. The term “medical treatment” was changed to “maternal glucose lowering medical treatment (metformin and/or insulin)”.
13) LINE 180: It would be helpful to use “FW” consistently (instead of spelling out fetal weight) throughout the manuscript.
Thank you for this comment. In the manuscript, we replaced “Fetal weight” by “FW”, “Fetal weight centile“ by “FWC”, and for a matter of consistency “Fetal abdominal circumference” by “FAC”,“ Fetal abdominal circumference centile” by “FACC”, “Neonatal weight” by “NW”, “Neonatal weight centile” by “NWC”, and “Gestational weight gain” by “GWG”. For the tables, we did not use abbreviations. For a matter of consistency, we also replaced “Abdominal circumference” by “Fetal abdominal circumference” in all tables.
14) LINES 179, 182, 184, 189, 190: Are the inverse associations indicated in Table 2 and Table 3? I believe they are reflected in Table A2.
The significant inverse associations between fetal anthropometric parameters with neonatal outcomes were observed for SGA and prematurity. They are shown in Table 2 and 3 as reduced OR (OR <1) for binary outcomes. Our data are shown as OR except for neonatal weight centile. To be clearer on the difference between beta-coefficients for continuous outcomes and OR for binary outcomes, beta-coefficients are now marked in the table (††, ‡‡, §§).
DISCUSSION
15) *What about HbA1c data?
LINES 312-322: Thank you for your interest in HbA1c. The following sentence was added in the discussion section “HbA1c at the last visit at the GDM clinic was not associated with neonatal outcomes, which may be related to a smaller sample size due to missing data.”
16) LINES 248-249: Is "Maternal medical treatment” referring to GDM-related treatment or any pregnancy-related treatment? I suspect GDM-related. If so, perhaps the term "GDM-related treatment" can be used?
LINES 268-269: “Maternal medical treatment” was changed to “Maternal glucose lowering medical treatment requirement”.
17) LINES 249-252: Did you mean, “... whereas abdominal 250 circumference centile WAS THE MOST POWERFUL PREDICTOR for prematurity, and abdominal circumference centile >90% for emergency 251 cesarean section”?
LINES 270-273: The phrase was changed as proposed to “FWC was found to be the most powerful predictor for NWC, LGA, and SGA, whereas FACC was the most powerful predictor for prematurity, and FACC >90% for emergency cesarean section.”
18) LINES 254-256: Did you mean, “...while fetal weight centile > 90% WAS AN INDEPENDENT PREDICTOR for neonatal birth weight, LGA, and 255 emergency cesarean section”?
LINES 276-277: The phrase was changed as proposed to “...while FWC > 90% was an independent predictor for neonatal birth weight, LGA, and emergency cesarean section.”
19) LINES 257, 267, “...a treated population...” Is this referring to women with GDM who managed their blood glucose with medication?
LINES 64, 278-279, 290: Thank you for addressing this point. Using the term “treated population” we refer to a population with a regular follow-up in a clinical setting. For a matter of clarity, we changed the phrase “and neonatal complications in the context of a treated population with GDM” by “and neonatal complications in the context of a population with GDM in a clinical setting”. In the lines 64 and 290 the word “treated” was eliminated.
20) LINES 283-286: Did you mean, "Fetal weight centile was found to be the most powerful fetal predictor for neonatal weight centile, LGA, and SGA, while abdominal circumference centile was superior for the prediction of prematurity AND abdominal circumference centile >90% WAS A STRONGER PREDICTOR for emergency cesarean section”?
LINES 307-310: The phrase was adapted according to your proposition to “FWC was found to be the most powerful fetal predictor for NWC, LGA, and SGA, while FACC was superior for the prediction of prematurity, and FACC >90% was a stronger predictor for emergency cesarean section.”
21) LINES 291-293: Did you mean, "In our study, fetal weight centile was more relevant for the prediction of neonatal anthropometric parameters at birth, whereas abdominal circumference WAS MORE RELEVANT for the prediction of other outcomes such as prematurity and emergency cesarean section."
LINES 314-317: The phrase was adapted according to your proposition to “In our study, FWC was more relevant for the prediction of neonatal anthropometric parameters at birth, whereas FACC was more relevant for the prediction of other outcomes such as prematurity and emergency cesarean section.”
CONCLUSIONS
22) *What about HbA1c data?
HbA1c has been discussed in the results and discussion section. Due to the fact that the aim of our study was to assess the utility of fetal anthropometric variables to predict the most relevant adverse neonatal outcomes in a clinical population with GDM beyond the known impact of maternal anthropometric and metabolic parameters, the impact of maternal parameters is not addressed in the conclusion.
23) LINES 312-315: Did you mean, "Fetal weight centile was found to be the most relevant predictor for neonatal anthropometric parameters (weight centile, LGA, SGA), WHEREAS abdominal circumference centile and abdominal circumference centile >90% WERE THE MOST RELEVANT PREDICTORS for prematurity and emergency cesarean section respectively."
LINES 340-343: The phrase was changed as proposed to “FWC was found to be the most relevant predictor for neonatal anthropometric parameters (weight centile, LGA, SGA) and FACC and FACC >90% the most relevant predictors for prematurity and emergency cesarean section respectively.”